# Covalent narlaprevir- and boceprevir-derived hybrid inhibitors of SARS-CoV-2 main protease

Daniel W. Kneller[1], Hui Li[2], Gwyndalyn Phillips [1], Kevin L. Weiss [1], Qiu Zhang[1], Mark A. Arnould [2], Colleen B. Jonsson[3,4,5], Surekha Surendranathan[5], Jyothi Parvathareddy[5], Matthew P. Blakeley [6], Leighton Coates [7], John M. Louis [8], Peter V. Bonnesen [2✉] & Andrey Kovalevsky [1✉]

Emerging SARS-CoV-2 variants continue to threaten the effectiveness of COVID-19 vaccines, and small-molecule antivirals can provide an important therapeutic treatment option. The viral main protease (M$^{pro}$) is critical for virus replication and thus is considered an attractive drug target. We performed the design and characterization of three covalent hybrid inhibitors BBH-1, BBH-2 and NBH-2 created by splicing components of hepatitis C protease inhibitors boceprevir and narlaprevir, and known SARS-CoV-1 protease inhibitors. A joint X-ray/neutron structure of the M$^{pro}$/BBH-1 complex demonstrates that a Cys145 thiolate reaction with the inhibitor's keto-warhead creates a negatively charged oxyanion. Protonation states of the ionizable residues in the M$^{pro}$ active site adapt to the inhibitor, which appears to be an intrinsic property of M$^{pro}$. Structural comparisons of the hybrid inhibitors with PF-07321332 reveal unconventional F···O interactions of PF-07321332 with M$^{pro}$ which may explain its more favorable enthalpy of binding. BBH-1, BBH-2 and NBH-2 exhibit comparable antiviral properties in vitro relative to PF-07321332, making them good candidates for further design of improved antivirals.

[1] Neutron Scattering Division, Oak Ridge National Laboratory, Oak Ridge, TN 37831, USA. [2] Center for Nanophase Materials Sciences, Oak Ridge National Laboratory, Oak Ridge, TN 37831, USA. [3] Department of Microbiology, Immunology and Biochemistry, University of Tennessee Health Science Center, Memphis, TN 38103, USA. [4] Institute for the Study of Host-Pathogen Systems, University of Tennessee Health Science Center, Memphis, TN, USA. [5] Regional Biocontainment Laboratory, The University of Tennessee Health Science Center, Memphis, TN 38105, USA. [6] Large Scale Structures Group, Institut Laue–Langevin, 71 Avenue des Martyrs, 38000 Grenoble, France. [7] Second Target Station, Oak Ridge National Laboratory, Oak Ridge, TN 37831, USA. [8] Laboratory of Chemical Physics, National Institute of Diabetes and Digestive and Kidney Diseases, National Institutes of Health, DHHS, Bethesda, MD 20892-0520, USA. ✉email: bonnesenpv@ornl.gov; kovalevskyay@ornl.gov

CoronaVIrus Disease 2019 (COVID-19), caused by the novel severe acute respiratory syndrome coronavirus 2 (SARS-CoV-2) continues to ravage the human population with wave after wave. The COVID-19 pandemic has had a catastrophic impact on human life and well-being, with nearly 500 million confirmed cases and over 6.1 million deaths to date globally and with millions of people suffering from long COVID complications[1–3]. While vaccines are now readily available in some countries to protect from the disease[4], a substantial proportion of the global population remains unvaccinated due to various factors, including an inability to vaccinate due to pre-existing conditions or unwillingness due to personal views. This situation promotes, at least in the near future, the selection of escape variants with the potential for multiple mutations and thus threatens the effectiveness of existing vaccines[5–7]. Small-molecule antiviral drugs targeting viral proteins, essential for the virus replication, can help alleviate this problem as the mechanism of action of such drugs would not be affected by mutations in the spike protein. However, repurposing of known clinical drugs for the treatment of COVID-19 has not met with much success[8–10]. Therefore, the design and development of specific antivirals against SARS-CoV-2 remain an urgent and pressing matter.

Among the initial steps in the virus replication cycle, the proteolytic function of the 3-chymotrypsin-like protease (3CL^pro, or main protease, M^pro) is vital for SARS-CoV-2 replication. After translation of the viral genomic mRNA by the host ribosomes, M^pro cuts the resulting large polyproteins pp1a and pp1ab into nonstructural proteins that assemble into the replication-transcription complex[11,12]. Thus, inhibition of the M^pro action impairs viral replication. Accordingly, there has been a significant increase in research activity over the past two years to create specific M^pro inhibitors with potential therapeutic applications[13–19]. Historically, the design, development, and clinical usage of HIV-1 and hepatitis C virus protease inhibitors[20,21] informs us that multiple continuously improving clinical drugs will be required to overcome various challenges when treating viral infections.

M^pro is a cysteine protease[22,23] that hydrolyzes peptide bonds by utilizing a noncanonical catalytic dyad comprised of Cys145 and His41, and an oxyanion hole formed by the main-chain amide NH groups of Gly143, Ser144, and Cys145 (Fig. 1a). The oxyanion hole evolved to stabilize the negatively charged tetrahedral intermediates along the M^pro-catalyzed reaction pathway. Furthermore, catalysis is believed to be assisted by a conserved water molecule that is hydrogen bonded to His41, His164, and Asp187, which together with Arg40 and Tyr54 comprise a partially negatively charged cluster thought to act as the third catalytic residue[24]. The M^pro active site can fit substrates or inhibitors containing chemical groups at positions P5 through P3′ to occupy substrate-binding subsites S5-S3′[25,26]. Subsites S1 and S2 are the most selective. S1 exclusively binds Gln, whereas S2 prefers medium-sized hydrophobic residues like Leu or Phe[13,27]. Two major strategies have been employed to inhibit M^pro efficiently. On the one hand, M^pro inhibitors can be noncovalent, binding to the enzyme active site only through hydrogen bonding and hydrophobic interactions. Conversely, the nucleophilic nature of Cys145 thiolate can be exploited by utilizing reactive electrophilic groups known as warheads that form an additional covalent bond with the enzyme improving inhibitor binding affinity relative to a noncovalent molecule. For example, several research groups have focused their attention on structure-based drug design of noncovalent M^pro inhibitors starting from the anti-epileptic drug perampanel[28,29], or from ML188 and ML300 previously designed to inhibit SARS-CoV-1 M^pro[30–33]. We have recently performed a structure-activity relationship study[34] on a noncovalent compound newly discovered by high-throughput virtual screening[35]. Others have designed and evaluated a series of peptidomimetic reversible or irreversible covalent inhibitors[36–38], or nonpeptidic small molecules with a novel chemical scaffold[39].

Hepatitis C virus clinical protease inhibitor boceprevir (Victrelis^TM, Merck)[40] and the feline peritonitis virus protease inhibitor GC-376 (Fig. 1b), both peptidomimetic reversible covalent inhibitors, demonstrated effective binding and inhibition of SARS-CoV-2 M^pro, and robust in vitro antiviral activity[41–44]. GC-376 is a prodrug that turns into the reactive aldehyde in aqueous solution. Not surprisingly, the chemical structures of boceprevir and GC-376 were combined to create hybrid inhibitors with improved in vitro and in vivo antiviral properties[45,46]. Researchers from Pfizer followed a similar strategy by additionally incorporating functional groups from compounds previously designed to inhibit SARS-CoV-1 M^pro[47,48]. Pfizer's strategy ultimately resulted in the design and development of an oral drug PF-07321332 (nirmatrelvir, Fig. 1b)[49] that has shown excellent efficacy in preventing hospitalization and death (https://clinicaltrials.gov/ct2/show/NCT04960202). Paxlovid^TM (PF-07321332/ritonavir) was recently approved by the U.S. Food and Drug Administration for emergency use to treat COVID-19 patients. Building from our recent hepatitis C virus protease inhibitor repurposing work[43] that included inhibitors boceprevir and narlaprevir (Arlansa^TM, R-Pharm, Russia[50,51]), we have followed an analogous strategy, working independently from and in parallel with other research groups, to design hybrid reversible covalent inhibitors of M^pro that combine the chemical features of boceprevir and narlaprevir with earlier SARS-CoV-1 M^pro inhibitors containing the widely used P1 γ-lactam and electrophilic arylketone or nitrile functionalities as warheads[47,48,52]. Our design logic also considered the observations and conclusions

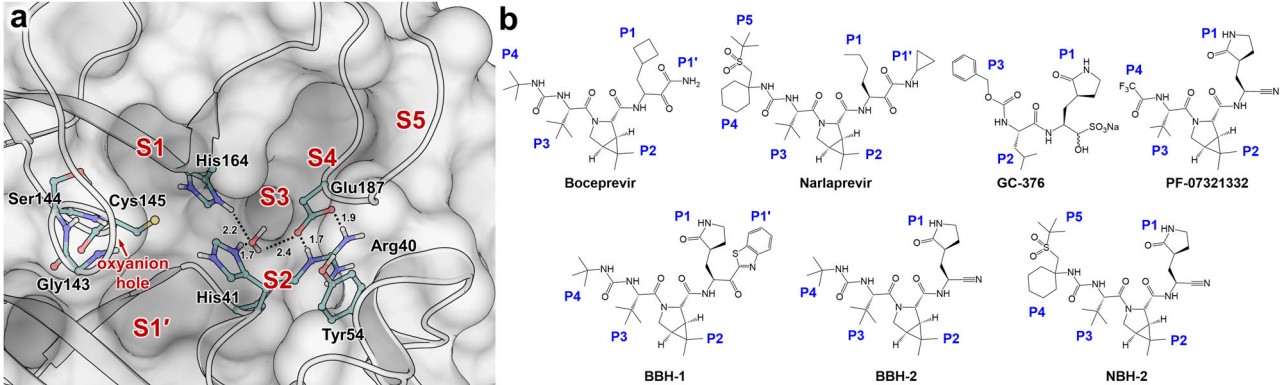

**Fig. 1 Active site architecture of SARS-CoV-2 M^pro and chemical structures of protease inhibitors. a** Substrate binding subsites are indicated as S5-S1′, including the oxyanion hole. **b** Reversible covalent inhibitors showing chemical groups P5-P1′ that bind in the subsites S5-S1′, respectively.

from our recent neutron structure of M$^{pro}$ in complex with another hepatitis C virus protease inhibitor, telaprevir[53]. In the M$^{pro}$/telaprevir complex, we found the oxygen of the hemithioketal formed by Cys145 reacting with the ketone warhead is protonated and makes a distorted, possibly weak, hydrogen bond with His41. Based on this observation, we hypothesized that an inhibitor's affinity might be increased by directing its warhead to form hydrogen bonding interactions with the oxyanion hole instead of His41.

Here, we present the design and characterization of three hybrid, reversible covalent M$^{pro}$ inhibitors named BBH-1, BBH-2, and NBH-2 (Fig. 1b). To map out the atomic details of inhibitors binding to the M$^{pro}$ active site, we determined a 2.2 Å joint X-ray/ neutron (XN) structure of M$^{pro}$/BBH-1 complex and 1.8 Å X-ray structures of M$^{pro}$/BBH-2 and M$^{pro}$/NBH-2 complexes at near-physiological temperature and pH. For direct comparison, we also obtained a 2.0 Å room-temperature X-ray structure of M$^{pro}$ in complex with PF-07321332. The M$^{pro}$/BBH-1 XN structure captured changes of the protonation states relative to the inhibitor-free M$^{pro}$[54] that could not be predicted a priori; in particular, the protonation states of four His residues located in the active site were directly visualized, revealing ionization states of His41, His163, His164, and His172. The XN structure also demonstrated that the hemithioketal generated by covalent binding of BBH-1 to Cys145 is not protonated, its oxygen being observed as an alkoxy anion. Inhibitors BBH-2 and NBH-2 effectively inhibited M$^{pro}$ in vitro, with the dissociation constants ($K_d$) of 26–30 nM measured by isothermal titration calorimetry (ITC), which compare favorably with the 7 nM $K_d$ for PF-07321332 ($K_i = 3.1$ nM measured by Owen et al.)[49]. Our study provides unique information on the structural and thermodynamic details of the hybrid reversible covalent inhibitors binding to SARS-CoV-2 M$^{pro}$ and demonstrates that these compounds are viable for further design of potent antivirals against SARS-CoV-2.

## Results

**Design of hybrid M$^{pro}$ inhibitors**. We initiated the design of hybrid inhibitors by careful examination of the chemical structures of boceprevir, narlaprevir, and telaprevir and their mode of binding to M$^{pro}$ based on our recent room-temperature X-ray structures[43]. These inhibitors belong to the class of reversible covalent inhibitors, each containing an electrophilic ketone warhead that Cys145 attacks to generate the covalent hemithioketal adduct. The inhibitors' chemical architectures are peptidomimetic, harboring mainly hydrophobic groups at positions P4-P1′ (P5- P1′ for narlaprevir, Fig. 1b). Importantly, X-ray crystallography provided evidence that these compounds have stereochemistry promoting their binding to M$^{pro}$[41–43]. We have also recently shown by neutron crystallography that in the M$^{pro}$/ telaprevir complex, the hemithioketal is protonated making a distorted, possibly weak, hydrogen bond with neutral His41. We hypothesized that directing the warhead into the oxyanion hole might benefit an inhibitor's binding affinity. We accomplished this by (a) substituting the ketoamide group of boceprevir with the keto-benzothiazole moiety where the aromatic substituent is P1′, or (b) introducing the nitrile warhead that fully eliminates the influence of a P1′ group. Next, we substituted the hydrophobic P1 groups of boceprevir and narlaprevir with a γ-lactam, an established mimic of P1 Gln, found in GC-376 and earlier inhibitors of SARS-CoV-1 M$^{pro}$[42,47,48]. A P1 group must have a hydrogen bonding acceptor capability to form a strong hydrogen bond with His163 in the S1 substrate binding subsite. Recent neutron crystallographic structures of M$^{pro}$ revealed that His163 adapts from the Nδ1-D tautomer to the protonated and

positively charged state in inhibitor complexes[34,53]. We therefore designed and synthesized the hybrid reversible covalent inhibitors BBH-1, BBH-2, and NBH-2 (Fig. 1 and Supplementary Fig. 1). We rescinded consideration of a narlaprevir-derived inhibitor with the keto-benzothiazole moiety due to the poor solubility of BBH-1.

**Adaptation of M$^{pro}$ active site electrostatics upon BBH-1 binding**. To gain insight into the atomic details of a hybrid inhibitor binding to M$^{pro}$, we obtained a 2.2 Å neutron crystallographic structure of the M$^{pro}$/BBH-1 complex jointly refined with 1.85 Å X-ray diffraction data. Both datasets were collected at room temperature and neutral pH from the same deuterated protein crystal (Supplementary Table 1). The joint XN structure revealed accurate locations of hydrogen atoms (observed as deuterium atoms, D), directly mapping the effect of the inhibitor binding on the active site electrostatics. For clarity, the hydrogen bonds observed in the XN structure are described by measuring distances between a D atom from a hydrogen bond donor and a heavy atom belonging to a hydrogen bond acceptor.

BBH-1 is presented in both electron and nuclear density maps (Fig. 2). The maps provide evidence that the Cys145 thiolate has reacted with the keto-warhead of BBH-1 to form a reversible covalent hemithioketal. According to the nuclear density (Fig. 3a), the hemithioketal oxygen is not protonated and thus is observed as an alkoxy anion (C–O$^-$). Its negatively charged oxygen is inserted into the oxyanion hole, and the charge is stabilized by a 1.9 Å hydrogen bond with the main-chain amide N-D of Cys145. The alkoxy anion is hydrated by a water molecule, which makes a 1.8 Å hydrogen bond with the charged oxygen. This water's lone electron pairs interact with the P1′ benzothiazole conjugated π-system through $n$-$π^*$ contacts with distances of 2.8–3.5 Å[55,56]. The benzothiazole nitrogen accepts a 1.7 Å hydrogen bond from another hydration water molecule that is also hydrogen-bonded to the Gly143 main-chain N-D with a distance of 1.9 Å. The latter water molecule is part of the water network that connects benzothiazole to the Asn142 side chain and a carbonyl group of BBH-1 (Figs. 2 and 3a).

The γ-lactam of the P1 group inserts into the substrate binding subsite S1 cavity formed by a stretch of residues from Phe140 to Asn142 on one side, and Glu166 and His172 on the other side (Fig. 3b). The S1 subsite is further capped by Ser1′ from the second M$^{pro}$ protomer, and P1 is secured by His163 at the bottom. The γ-lactam 5-membered ring forms a conventional envelope conformation. The carbonyl oxygen makes a short 1.7 Å hydrogen bond with the protonated imidazolium side chain of His163. A similar hydrogen bond was observed between the P1 uracil of a noncovalent inhibitor Mcule-5948770040 and His163[34]. However, unlike the uracil group, the γ-lactam doesn't make a C-H...O interaction with His172. Although the γ-lactam's amide N-D is within hydrogen bonding distance (2.1 Å) to the Glu166 carboxylate, the geometry of the contact is significantly distorted from the ideal hydrogen bond geometry. The N-D...O angle is 147°, and the N-D bond vector is almost perpendicular to the carboxylate plane. Moreover, the Glu166 side chain is already hydrogen bonded with His172 (2.2 Å) and the N-terminal ND$_3^+$ of Ser1′ (1.9 Å). This suggests that the γ-lactam...Glu166 contact is probably weak. Nonetheless, the Glu166 carboxylate rotates by about 90° from its observed position in the inhibitor-free M$^{pro}$ structure. It forms a new hydrogen bond with the neutral His172, as was observed in our other inhibitor-bound M$^{pro}$ structures[34,43,53]. P2, P3, and P4 groups of BBH-1 are hydrophobic and do not form hydrogen bonds with M$^{pro}$. Conversely, the amide groups connecting P1/P2 and P2/P3 and the substituted urea moiety make several hydrogen bonds of

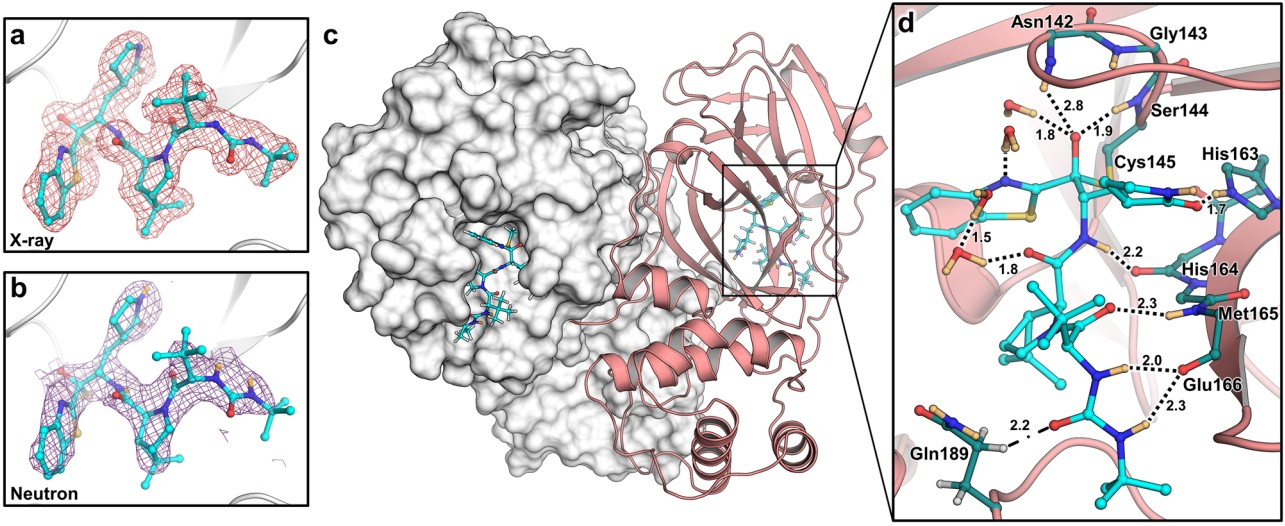

**Fig. 2 Joint X-ray/neutron crystal structure of deuterated SARS-CoV-2 M^pro with BBH-1. a** Electron and **b** nuclear density for BBH-1 **c** covalently bound in the M^pro active site. **d** Direct and water-mediated hydrogen bond interactions between BBH-1 and M^pro. Protein homodimer represented with one protomer as white surface and the other as salmon cartoon. Electron (pink mesh) and nuclear (purple mesh) density of BBH-1 (cyan ball and sticks) contoured to 1 σ. Hydrogen bonds are represented as black dashes while C-H···O interaction with Gln189 is represented as dash-dots, with distances in angstroms (Å). Deuterium atoms are colored in light orange.

2.0–2.3 Å with the main chains of His164 and Glu166. In addition, the carbonyl of the urea moiety forms an unconventional C-H…O interaction with the side chain of Gln189[57,58].

The P1′ benzothiazole is located in the substrate-binding subsite S1′ flanked by Thr25, His41, Cys44, and Met49 (Fig. 3c). Because of its steric bulkiness, the benzothiazole pushes His41 away from its position in the inhibitor-free M^pro structure[54], forcing the His41 side chain to rotate ~56° around the Cα-Cβ bond (Fig. 3d). The His41 movement causes the catalytic water molecule ($D_2O_{cat}$) to be ejected from the active site and His164 to become deprotonated at the Nδ1 atom. A similar cascade of structural changes was previously detected in the X-ray structures of other inhibitors containing the keto-benzothiazole moiety[19,49]. Interestingly, the His41 side chain imidazole does not flip 180° in the present structure and only shifts ~0.4 Å at Cα to accommodate the benzothiazole substituent, which creates an unconventional 2.1 Å C-H…N interaction with His164 (Fig. 3c). In the published X-ray structures[19,49], the His41 imidazole side chain was modeled in the flipped conformation, suggesting the formation of an N-H…N hydrogen bond with His164 (Supplementary Fig. 2). In both M^pro/BBH-1 and inhibitor-free M^pro XN structures, Nε2 of His164 donates its D in a hydrogen bond with the Thr175 hydroxyl, which forms a hydrogen bond with the main-chain carbonyl of Asp176. This hydrogen bond is conserved in all XN structures we have determined[34,43,53]. The neutral protonation state of His172 is also maintained because it is secured by the hydrogen bond with main-chain amide N of Gly138. Moreover, by altering the protonation states of His residues, the M^pro active site electrostatics adapts to the binding of BBH-1. Still, it maintains the overall charge of +1, as seen in the inhibitor-free M^pro [54].

**Binding of inhibitors containing nitrile warhead**. To generate a detailed structural picture of the nitrile warhead-containing hybrid inhibitors BBH-2 and NBH-2 binding to M^pro, we performed room-temperature X-ray crystallographic analysis of M^pro/BBH-2 and M^pro/NBH-2 complexes at 1.8 Å resolution. For direct comparison, we also determined a room-temperature X-ray structure of M^pro in complex with PF-07321332 to a resolution of

2.0 Å (Supplementary Table 2). The hydrogen bond distances in the X-ray structures are given below with the distances measured between the heavy atoms because, unlike in the M^pro/BBH-1 XN structure, hydrogen atoms are not observed in the electron density at such resolutions.

Both BBH-2 and NBH-2 bind in an essentially identical fashion to the M^pro active site. The nitrile warhead forms the covalent thioimidate ester conjugate with the catalytic Cys145 and is inserted into the oxyanion hole (Fig. 4). The thioimidate N forms a 3.0 Å hydrogen bond with the main-chain amide NH of Cys145 but is too far from Gly143 to form a hydrogen bond (N…N distance is >3.6 Å) and faces the Ser144 main chain almost perpendicularly. In addition, the thioimidate N is hydrated by a water molecule. The interactions are similar to those for the hemithioketal alkoxy anion of BBH-1. It, therefore, appears that in the case of these inhibitors only the main chain of Cys145, and not of Gly143 or Ser144, is involved in hydrogen bonding with the covalently modified warhead. Whether the thioimidate N is protonated/neutral or deprotonated/negatively charged as observed for BBH-1 hemithioketal cannot be discerned from the X-ray structure. As observed in the M^pro/BBH-1 XN structure, BBH-2 and NBH-2 make several hydrogen bonds with the main chain atoms of His164 and Glu166 with the distances of 3.0-3.2 Å, and a 2.7 Å hydrogen bond with the side chain of His163. Of note, the thioimidate N of PF-07321332 is farther away from Cys145 main-chain amide N at 3.4 Å, but its P1 group and the amide connecting P3 and P4 form slightly shorter hydrogen bonds of 2.6 and 2.8 Å with His163 and Glu166, respectively, compared to those seen for BBH-2 or NBH-2. Other interactions made by BBH-2, NBH-2 and PF-07321332 with M^pro are hydrophobic except those made by the P4 $CF_3$ group of PF-07321332. The $CF_3$ substituent makes 2.8 and 3.2 Å contacts with the main-chain carbonyl oxygens of Glu166 and Thr190. Such F…O interactions have previously been characterized in small-molecule compounds as attractive multipolar interactions[59].

We have previously demonstrated[43,60] that the active site cavity of M^pro exhibits a significant degree of conformational plasticity and malleability, which assists in the binding of sterically bulky inhibitors. Specifically, the most malleable regions of the active

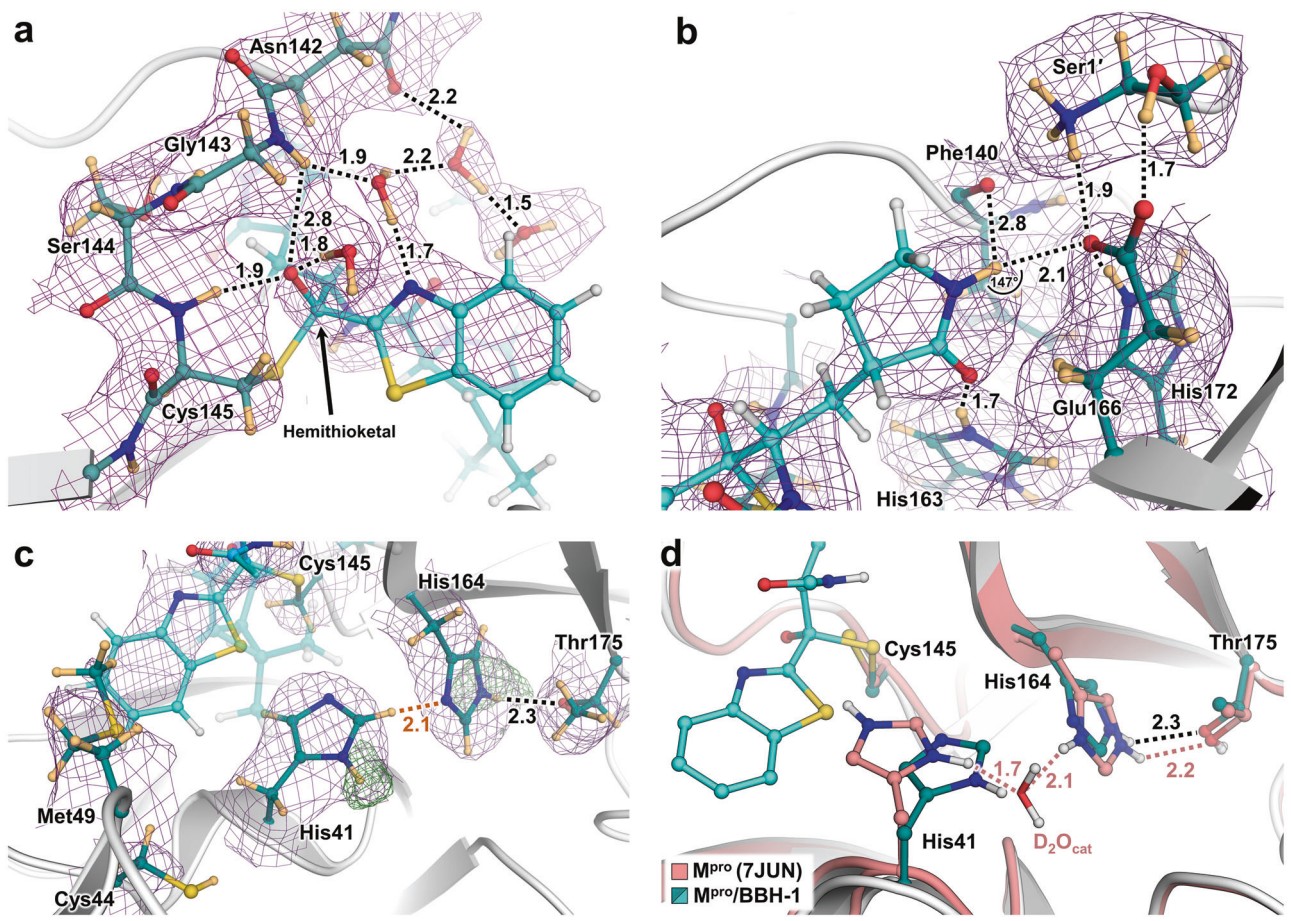

**Fig. 3 Direct observation of BBH-1 electrostatic interactions with M^pro determined by neutron crystallography. a** Cys145 reacts with the keto-benzothiazole warhead of BBH-1 to form a covalent hemithioketal group with an alkoxy anion directed towards the oxyanion hole and hydrogen bonding to an accompanied D₂O molecule. **b** Hydrogen bond network involved with the P1 γ-lactam of BBH-1 in the M^pro S1 pocket. **c** The catalytic His41 in M^pro/BBH-1 is singly protonated on Nδ1 and forms a C-D···N interaction (orange dashes) with His164. **d** The P1' benzothiazole group of BBH-1 sterically swings His41 from its position in the ligand-free XN structure of M^pro (7JUN) and evicts the catalytic water molecule. Protonation states of His41 and His164 change from doubly protonated to singly protonated. BBH-1 as cyan ball and sticks. Ligand-free M^pro colored in salmon. Neutron 2Fo-Fc map (purple mesh) contoured at 1.0 σ. Omit nuclear Fo-Fc for D atoms on His41 and His164 are shown as green mesh contoured at 4 and 2.8 σ respectively. Hydrogen bonds are represented as black dashes. Distance is given in angstroms (Å). Deuterium atoms are colored in light orange.

site cavity are the S2 helix (residues 46–51), S4 β-hairpin loop (residues 165–170) and S5 loop (residues 189–194). These regions can shift by as much as 2.5 Å relative to their positions in the inhibitor-free structure[43]. It is, therefore, instructive to compare the shapes active site cavities adopt in the M^pro complexes reported here with that in the inhibitor-free enzyme. Accordingly, we superimposed the four studied complexes over the XN structure of inhibitor-free M^pro[54]. Although the overall root-mean-square deviations (RMSDs) are low, ~0.4 Å for all structures, significant changes in the conformations of the regions directly interacting with the inhibitors are apparent (Fig. 5). The S2 helix and S4 β-hairpin loop shift by as much as 2 Å, whereas the S5 loop moves by about 1.7 Å in M^pro/BBH-1 to support the induced fit of the inhibitor. The conformational changes are less drastic in M^pro/BBH-2 and M^pro/NBH-2 for the S4 β-hairpin loop (~1.7 Å) and S5 loop (~1.5 Å) but are more pronounced for the S2 helix (~2.5 Å), although P2 groups are identical in these inhibitors and the P4-P5 groups of NBH-2 are sterically substantially bulkier than the *tert*-butyl substituent of BBH-1 or BBH-2. In this regard, a noteworthy observation is that in the M^pro/PF-07321332 complex, the conformational changes of the S2 helix and S4 β-hairpin loop are very similar to those seen in the other two nitrile warhead-containing inhibitors. This finding

is expected for the S2 helix because the P2 groups of the inhibitors are identical. However, the P4 CF₃ group of PF-07321332 is sizably smaller than the *tert*-butyl moiety of BBH-2 or the cyclohexyl group of NBH-2. The S4 β-hairpin loop shift may partially be caused by electrostatic repulsion with the highly electronegative CF₃ group. Conversely, the S5 loop moves by just ~1 Å in M^pro/PF-07321332 relative to the inhibitor-free M^pro in agreement with the reduced steric size of the CF₃ group and the lack of a P5 group present in NBH-2. Overall, the above structural analysis reinforces the idea that the M^pro active site cavity can accommodate various chemical groups by induced fit, especially in the subsites S4 and S5. The room-temperature structure of the M^pro/PF-07321332 complex superimposes onto the 100 K structure of this complex reported previously[49] with an RMSD of 0.5 Å (Supplementary Fig. 3). It appears, however, that the M^pro active site is more open at room temperature. The distances between the S2 helix and the S4 β-hairpin loop, and between the S5 loop and the S4 β-hairpin loop increase by ~0.7 Å in the room-temperature structure relative to those in the 100 K structure. This could be due to differences in packing interactions between the unit cell types and the different data collection temperatures, but the change underscores the very flexible nature of the enzyme's active site.

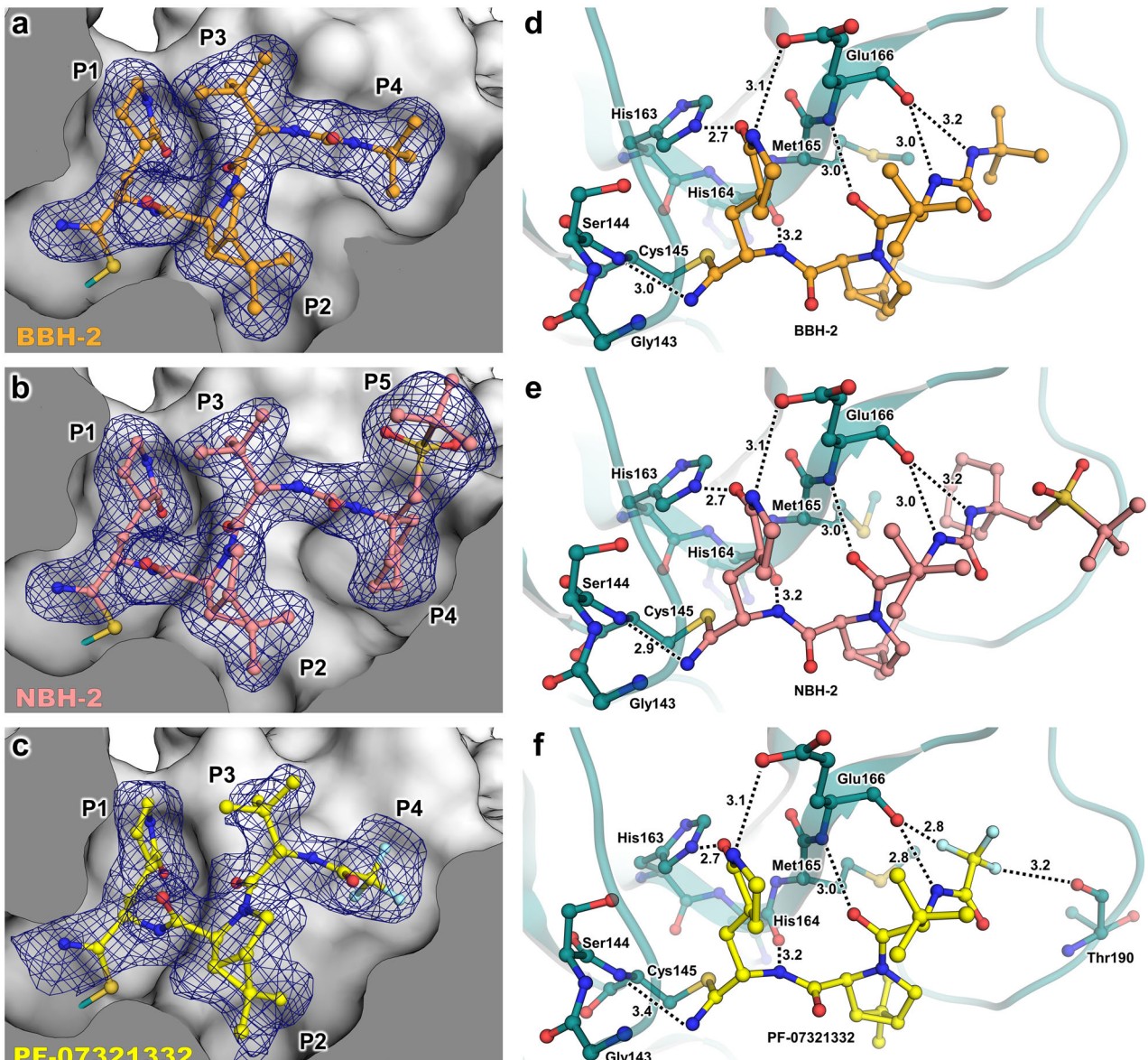

**Fig. 4 Room-temperature X-ray crystal structures of M$^{pro}$ in complex with nitrile warhead inhibitors BBH-2, NBH-2, and PF-037321332. a–c** Inhibitors BBH-2, NBH-2, and PF-037321332 bind to the active site of M$^{pro}$ and form a covalent thioimidate with Cys145. Polder omits maps of inhibitor atoms (blue mesh) contoured to 3σ. **d–f** Hydrogen bonds between inhibitors and M$^{pro}$ are shown as black dashes with distances in angstroms (Å).

**Binding thermodynamics of the inhibitors**. To determine the dissociation constants ($K_d$) and assess the binding thermodynamics of nitrile warhead-containing inhibitors BBH-2, NBH-2, and PF-07321332, we performed isothermal titration calorimetry (ITC, Table 1 and Supplementary Fig. 4). We could not carry out ITC measurements for BBH-1 due to insufficient solubility of the inhibitor in buffer at the concentrations required for ITC even in the presence of 1.5% DMSO. BBH-2 and NBH-2 demonstrated high binding affinity to M$^{pro}$, with almost identical $K_d$ values of 0.026 ± 0.016 and 0.030 ± 0.007 μM. Moreover, PF-07321332 binds to M$^{pro}$ with several fold lower $K_d$, demonstrating its superior binding affinity compared to the other two inhibitors and in agreement with the $K_i$ of 0.0031 μM reported previously[49]. Binding of these inhibitors to M$^{pro}$ is driven largely by enthalpy. Notably, ΔH and ΔS of binding for BBH-2 and NBH-2 are virtually identical, even though their P4 groups are different, and BBH-2 lacks a P5 group. This indicates that the NBH-2's P5 group has no measurable effect on its binding to

M$^{pro}$. The ΔH of binding for PF-07321332 is more favorable than those for BBH-2 and NBH-2 by about −2 kcal/mol, implying that P4 CF$_3$ group makes better interactions with M$^{pro}$ amino acid residues compared to the hydrophobic groups on the other two inhibitors. The improvement in ΔH of binding for PF-07321332, however, is somewhat compensated by the less favorable ΔS of binding relative to BBH-2 and NBH-2. The difference in the ΔS of binding could be due to greater entropy of dehydration, which is required for the bulky hydrophobic P4 and P5 groups in BBH-2 and NBH-2 to bind to M$^{pro}$ compared to that for the small CF$_3$ group present in PF-07321332. Nevertheless, PF-07321332 binds to the enzyme with a very small change in entropy, akin to the binding of the HL-3 series of noncovalent inhibitors[34]. Consequently, although PF-07321332 possesses a higher binding affinity than those of BBH-2 and NBH-2, the interplay of ΔH and ΔS components manifests in the free energy of binding (ΔG) for PF-07321332 being only −0.7 kcal/mol more favorable than ΔG of binding for the other two inhibitors.

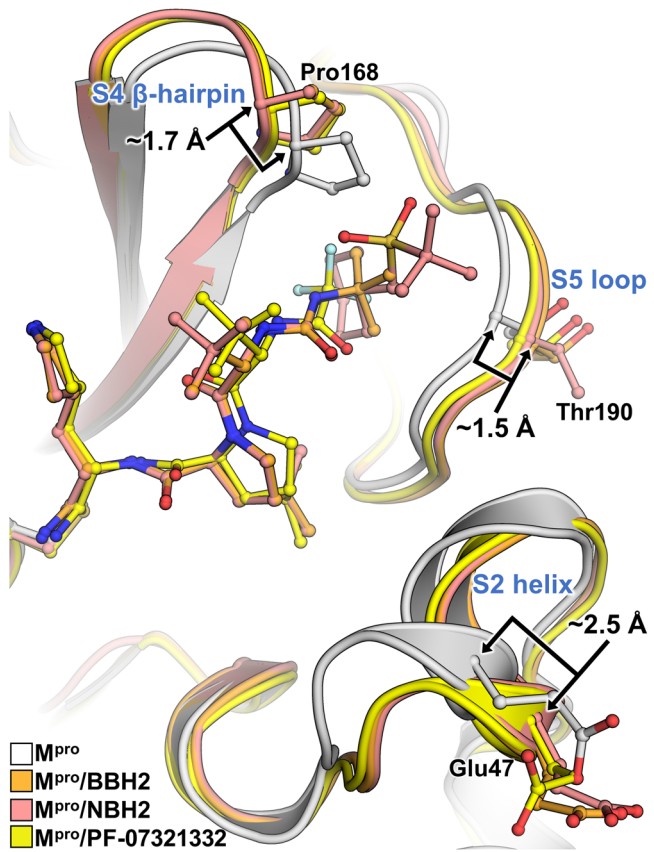

**Fig. 5 Active site structural plasticity of M^pro bound to nitrile warhead inhibitors.** Comparison of ligand-free M^pro (7JUN, white cartoon) with the BBH-2, NBH-2, and PF-037321332 complexes shows significant structural shifts in protein subsites S2, S4, and S5 responsible for binding to inhibitor P2, P4, and P5 moieties respectively. Distances are given in angstroms (Å). Superposition is calculated by the least-square-fitting of Cα atoms.

**Antiviral activity.** The in vitro antiviral activity of all four compounds, BBH-1, BBH-2, NBH-2, and PF-07321332, was evaluated in Vero E6 TMPRSS cells. These cells express significant amounts of P-glycoprotein that acts as an efflux pump capable of efficiently removing these compounds from the cytoplasm[49,61]. Therefore, we assessed the antiviral properties of the inhibitors in the absence and presence of CP-100356, a P-glycoprotein inhibitor. In the absence of CP-100356, BBH-1, BBH-2, and NBH-2 inhibited viral replication rather weakly, with EC$_{50}$s in the 14–16 μM range, whereas PF-07321332 showed a more potent EC$_{50}$ of 0.88 μM (Table 2, Supplementary Fig. 5). These EC$_{50}$ values for our compounds are similar to those of boceprevir published previously[41,42]. The addition of 2 μM CP-100356 significantly improved the antiviral potency of all the compounds, with the EC$_{50}$ values decreasing at least 10-fold for BBH-1 (1.54 μM), BBH-2 (0.88 μM), and NBH-2 (1.82 μM). In the presence of CP-100356, PF-07321332 inhibited SARS-CoV-2 replication with EC$_{50}$ of 0.25 μM, which is consistent with the recently reported measurements[49]. No cytotoxicity was detected for all four inhibitors at concentrations up to 10 μM.

## Discussion

The current design approach for small-molecule inhibitors of SARS-CoV-2 M^pro has yielded an orally bioavailable compound PF-07321332 (nirmatrelvir, Paxlovid™, Pfizer Inc.), a reversible covalent peptidomimetic inhibitor that takes advantage of the nucleophilic properties of the catalytic Cys145 thiolate and

whose chemical structure combines features of boceprevir, GC-376 and other earlier inhibitors of SARS-CoV-1 M^pro. Further design of novel M^pro inhibitors, including covalent and non-covalent compounds, will undoubtedly be required to battle COVID-19 and to prepare for the possibility of future coronavirus outbreaks. Improved inhibitor design should consider employing both conventional hydrogen bonds and unconventional intermolecular interactions, such as C-H···O(N) and X-H···π contacts, where X = O, N, S, or C[62], with M^pro residues. Thus, hydrogen atoms that represent about half of the total atom count in both small molecules and proteins cannot be overlooked. We have been using neutron protein crystallography to create a map of hydrogen atom positions and their movements that alter active site electrostatics due to inhibitor binding in SARS-CoV-2 M^pro[34,53,54]. This led us to the design of hybrid covalent inhibitors BBH-1, BBH-2, and NBH-2 with >100-fold improved affinity to M^pro, when compared to the IC$_{50}$ values of boceprevir (3.1–8.0 μM) and narlaprevir (4.7–5.1 μM) determined previously[41–43], whose chemical structures were the starting points in our design approach. These inhibitors together with PF-07321332 also exhibited robust antiviral properties, inhibiting SARS-CoV-2 replication in our assays at the low-to-sub μM concentrations.

The most important observations in the XN structure of M^pro/BBH-1 are the protonation states and hydrogen bonding contacts visualized in the oxyanion hole and S1 substrate binding subsite. As expected from the mechanism of cysteine protease-catalyzed peptide bond hydrolysis, the hemithioketal oxygen of BBH-1 was found not to be protonated. Its negative electric charge is stabilized by a hydrogen bond with the main chain amide N-H of Cys145 (Fig. 3). The P1 γ-lactam makes a strong hydrogen bond with His163 that becomes protonated and positively charged when BBH-1 binds to M^pro. This hydrogen bond is essential for inhibitor binding affinity, as its disruption leads to a significant loss in inhibitor potency, as was shown in our structure-activity relationship study on a noncovalent inhibitor[34]. Similar protonation states and hydrogen bond interactions can be envisaged in the M^pro complexes with the three nitrile warhead inhibitors studied here, in which the thioimidate nitrogen would be negatively charged. Protonation of His163 to gain a positive charge upon binding of an inhibitor irrespective of the P1 group's chemical nature is evidently an intrinsic property of this residue's side chain, a conclusion drawn based on our observations of this phenomenon in all neutron structures that we have determined so far[34,53,54]. Moreover, another remarkable property of the M^pro active site that appears to be invariant is its ability to maintain overall +1 electric charge when inhibitors bind through switching protonation states of His41, Cys145, His163, and His164 (Supplementary Table 3).

The binding of the nitrile warhead-containing inhibitors BBH-2, NBH-2, and PF-07321332 is enthalpically driven, an important property for a successful drug molecule. Although the three inhibitors are similar, the entropy of binding for the two former inhibitors is significant, whereas it is small for PF-07321332 and is substantially compensated by a more favorable enthalpy of binding. This makes PF-07321332 a superior inhibitor among the three compounds studied with ITC. This result is probably caused by substituting the bulky hydrophobic P4/P5 groups of BBH-2 and NBH-2 with a small electronegative CF$_3$ group that minimizes the hydrophobic effect and is also capable of forming favorable unconventional F···O interactions with M^pro. Thus, the CF$_3$ group conveys more favorable antiviral properties (Table 2) and was also shown to be beneficial for the compound's metabolic stability and pharmacokinetics[49]. Future M^pro inhibitor design can consider altering the P3 *tert*-Bu group that is exposed to solvent and makes no contacts with M^pro residues.

**Table 1 Binding affinities of nitrile warhead-containing inhibitors BBH-2, NBH-2, and PF-07321332 determined by isothermal titration calorimetry ($K_d$ and thermodynamic parameters $\Delta H$, $\Delta S$, and $\Delta G$ of binding).**

| Compound | $K_d$ (µM) | Stoichiometry (N) | $\Delta H$ (kcal mol$^{-1}$) | $\Delta S$ (cal mol$^{-1}$ K$^{-1}$) | $\Delta G$ (kcal mol$^{-1}$) |
|---|---|---|---|---|---|
| BBH-2 | 0.030 ± 0.007 | 1.000 ± 0.005 | −8.74 ± 0.08 | 5.40 | −10.4 |
| NBH-2 | 0.026 ± 0.016 | 0.990 ± 0.009 | −8.76 ± 0.17 | 5.63 | −10.5 |
| PF-07321332 | 0.007 ± 0.003 | 0.990 ± 0.003 | −10.75 ± 0.70 | 1.57 | −11.2 |
| GC-376 | 0.15 ± 0.03[a] | 0.99 ± 0.01 | −6.7 ± 0.1 | 9.1 | −9.4 |

[a]ITC values for GC-376 are taken from Nashed et al.[88].

**Table 2 In vitro antiviral parameters for SARS-CoV-2 M$^{pro}$ inhibitors measured in Vero E6 TMPRSS cells in the absence or presence of P-glycoprotein inhibitor CP-100356.**

| Compound | EC$_{50}$, µM (without CP-100356) | EC$_{50}$, [95% CI], µM (with CP-100356) | CC$_{50}$, µM |
|---|---|---|---|
| BBH-1 | 16.1 | 1.54 [0.82, 2.84] | >10 |
| BBH-2 | 15.4 | 0.88 [0.46, 1.66] | >10 |
| NBH-2 | 13.9 | 1.82 [0.90, 3.66] | >10 |
| PF-07321332 | 0.88 | 0.25 [0.07, 0.71] | >10 |

In summary, we designed three hybrid reversible covalent inhibitors of SARS-CoV-2 M$^{pro}$ based on the chemical structures of hepatitis C clinical protease inhibitors narlaprevir and boceprevir and compared them to the recently FDA-approved inhibitor PF-07321332. Our study combined chemical synthesis, neutron and X-ray crystallography at nearly physiological temperature, and in vitro measurements to obtain atomic details of the inhibitors' binding to M$^{pro}$, their binding thermodynamics and antiviral potency. We directly visualized an oxyanion bound in the catalytic site of M$^{pro}$, the negatively charged oxygen being generated when Cys145 reacts with the ketone warhead of the BBH-1 inhibitor. BBH-1, BBH-2, and NBH-2 showed significant antiviral activity that was only slightly lower than that for PF-07321332. Our results shed light on the atomic details of inhibitor binding to SARS-CoV-2 M$^{pro}$ and provide insights for future drug design.

## Methods

**General information**. Ni-NTA columns were purchased from Cytiva (Piscataway, New Jersey, USA). His-tagged Human Rhinovirus (HRV) 3C protease was purchased from Sigma (MilliporeSigma, St. Louis, MO). Crystallization reagents and supplies were purchased from Hampton Research (Aliso Viejo, California, USA). Crystallographic supplies for crystal mounting and X-ray and neutron diffraction data collection at room temperature were purchased from MiTeGen (Ithaca, New York, USA) and Vitrocom (Mountain Lakes, New Jersey, USA). PF-07321332 (CAS # 2628280-40-8) was purchased from MedChemExpress (Monmouth Junction, New Jersey, USA)

**Synthesis of BBH-1, BBH-2, and NBH-2**. Boceprevir and narlaprevir both possess alkyl substituted 3-amino-2-oxopropanamide endgroups. It was our intent to replace those moieties with P1 fragments containing the γ-lactam group and the desired nitrile or keto-benzothiazole warhead. To prepare the boceprevir-derived hybrids BBH-1 and BBH-2, the boceprevir fragment lacking the 3-amino-4-cyclobutyl-2-oxobutanamide moiety was utilized. This fragment, (1R,2S,5S)-3-((S)-2-(3-(tert-butyl)ureido)-3,3-dimethylbutanoyl)-6,6-dimethyl-3-azabicyclo[3.1.0]hexane-2-carboxylic acid, was coupled to either (S)-3-((S)-2-amino-3-(benzo[d]thiazol-2-yl)-3-oxopropyl)pyrrolidin-2-one (for BBH-1), or (S)-2-amino-3-((S)-2-oxopyrrolidin-3-yl)propanenitrile (for BBH-2), using standard amide forming methodologies employed in peptide coupling reactions. Similarly, to prepare NBH-2, the narlaprevir fragment (1R,2S,5S)-3-((S)-2-(3-(1-((tert-butylsulfonyl)methyl)cyclohexyl)ureido)-3,3-dimethylbutanoyl)-6,6-dimethyl-3-azabicyclo[3.1.0]hexane-2-carboxylic acid was coupled with (S)-2-amino-3-((S)-2-oxopyrrolidin-3-yl)propanenitrile (see Supplementary Fig. 1). All purchased reagents were at least 95% pure and the details of the syntheses are described in the Supporting Information.

**Expression and purification of hydrogenated and partially deuterated M$^{pro}$**. The SARS-CoV-2 M$^{pro}$ (NSP5) gene sequence was codon optimized before being cloned into a plasmid harboring a kanamycin resistance cassette (pD451-SR, Atum, Newark, CA). The protein construct features M$^{pro}$ flanked upstream by maltose binding protein (MBP) and downstream by a His6 tag[60]. A native N-terminus is attained during expression through an M$^{pro}$ autoprocessing site corresponding to the cleavage between NSP4 and NSP5 in the viral polyprotein, SAVLQ ↓ SGFRK, where ↓ denotes the cleavage site. A native C-terminus is produced through an HRV-3C protease cleavage site (SGVTFQ ↓ GP). Hydrogenated M$^{pro}$ was expressed in Escherichia coli and purified according to the established procedures[60,63]. Partially deuterated M$^{pro}$ was expressed using a bioreactor and purified as described recently[53]. Protein yields for hydrogenated M$^{pro}$ preparations averaged ~3.5 mg per 1 g cells and were used fresh or stored at −30 °C. Final yield for the partially deuterated purification was ~1.6 mg per 1 g of cell paste and was used immediately for crystallization. Similar to the above strategy, a second construct was used to express and purify the wild-type M$^{pro}$. Expression and purification were carried out as described previously[13,34]. This second source of M$^{pro}$ was used to determine the binding constants by ITC.

**M$^{pro}$-inhibitor complex crystallization**. Instructions for reproducing large-volume plate-shaped crystals of hydrogenated and partially deuterated M$^{pro}$ enzyme are accessible[63]. Discovery of conditions for crystallizing M$^{pro}$ was accelerated by automated high-throughput screening at the Hauptman-Woodward Medical Research Institute (HWI)[64]. Crystal aggregates of ligand-free M$^{pro}$ were reproduced locally and then converted into microseeds for growing single crystals. We recently discovered that microseeded M$^{pro}$ crystals grow exclusively in a cuboid morphology when co-crystallized with noncovalent ligand Mcule-5948770040 and its derivatives[34]. Cube-shaped crystals of M$^{pro}$ for current experiments were obtained by seed streaking from cube-shaped M$^{pro}$/Mcule-5948770040 co-crystals using a cat whisker or seeding tool.

Hydrogenated protein in 20 mM Tris, 150 mM NaCl, and 1 mM TCEP (tris(2-carboxyethyl)phosphine) pH 8.0 was concentrated to ~5–7 mg/mL for growing ligand-free crystals and co-crystals. Stocks of inhibitors were prepared at 50 mM in 100% dimethyl sulfoxide (DMSO) for crystallization purposes and stored at −30 °C. For co-crystallization, M$^{pro}$ was mixed with BBH-1, BBH-2, or NBH-2 at 1:5 molar ratio and allowed to incubate at room temperature for a minimum of 30 minutes before setting up crystal trays. Crystals grown for room-temperature X-ray diffraction used sitting drop vapor diffusion methodology with 18–21% PEG3350, 0.1 M Bis-Tris pH 6.5 (1 mL) as the precipitant solution. Crystallization drops of 20 µL at 1:1 ratio were seed struck with cubes as described above. Crystals appeared after 3 days of incubation at 14 °C and continued to grow to a typical volume of ~0.1 mm$^3$. Co-crystallization attempts with PF-7321332 failed, prompting crystal soaking. A pre-grown ligand-free M$^{pro}$ crystal was transferred to a different drop supplemented with PF-7321332 to a final concentration of 1.5 mM and 3% DMSO and allowed to soak for 10 min at room temperature before X-ray data collection. Crystals were mounted in MiTeGen (Ithaca, NY) room-temperature capillary setups for data collection.

For joint XN crystallography, partially deuterated M$^{pro}$ at 10.6 mg/mL was mixed with BBH-1 at a 1:5 molar ratio, incubated at room temperature for 30 min, then filtered through a 0.2 µm centrifugal filter. A Hampton 9-well sandwich box was set up with 240 µL drops at a 1:1 ratio of protein to 20% PEG3350, 0.1 M Bis-Tris pH 6.5 reservoir solution of 50 mL. Due to the favorable 3D shape for neutron data collection, nucleation in the cube-morphology was encouraged by seed streaking. After 19 days of incubation at 10 °C, a crystal measuring ~1.2 × 1 × 0.4 mm (0.5 mm$^3$) was mounted in a quartz capillary (Supplementary Fig. 6). The crystal was accompanied by 22% PEG3350 prepared in 100% D$_2$O to exchange labile hydrogens during the 2 weeks preceding the neutron data collection. A separate crystal from the same drop was mounted inside a glass capillary in the same fashion for room-temperature X-ray data collection used in joint X-ray/neutron refinement. The final pH of the crystallization drop was measured by a microelectrode to be 6.6, corresponding to a neutral pD of 7.0 (pD = pH + 0.4). This pD is identical to previously characterized neutron crystal structures of inhibitor-free M$^{pro}$, the M$^{pro}$/telaprevir covalent complex, and the M$^{pro}$/Mcule-5948770040 noncovalent complex[34,53,54].

**Room temperature X-ray diffraction data collection and structure refinement**. All room temperature X-ray crystallographic data was collected with a Rigaku HighFlux HomeLab instrument equipped with a MicroMax-007 HF X-ray generator, Osmic VariMax optics, and a DECTRIS Eiger R 4 M hybrid photon counting detector. X-ray diffraction data were integrated using the CrysAlis Pro software suite (Rigaku Inc., The Woodlands, TX) then reduced and scaled using Aimless[65] from the CCP4 suite[66]. Structures were solved by molecular replacement using Phaser[67] from CCP4 using PDB code 7N8C[34]. Each model was iteratively refined with Phenix.refine from the PHENIX suite[68,69] and COOT[70,71]. Geometry validation was aided by Molprobity[72]. All ligand restraints were generated with eLBOW[73] using geometry optimized by quantum mechanical calculations in Gaussian 16 at B3LYP/6-31 g(d,p) level of theory[74]. Final data collection and refinement statistics can be found in Supplementary Table 2.

**Neutron diffraction data collection**. Neutron diffraction was tested at room temperature on the IMAGINE[75–78] instrument located at the High Flux Isotope Reactor (Oak Ridge National Laboratory) using the broad bandpass functionality with neutron wavelengths between 2.8 and 10 Å. Neutron quasi-Laue diffraction data from a 0.5 mm$^3$ crystal of the M$^{pro}$/BBH-1 complex were collected at room temperature using the LADI-III diffractometer at the Institut Laue-Langevin (ILL) in Grenoble[79]. A neutron wavelength range ($\Delta\lambda/\lambda = 30\%$) of 2.80–3.75 Å was used for data collection with diffraction data extending to 2.2 Å resolution. The crystal was held stationary at different $\varphi$ (vertical rotation axis) for each exposure. A total of 22 images were recorded with an average exposure time of 23 h per image from three different crystal orientations. The neutron data were processed using the Daresbury Laboratory LAUE suite program *LAUEGEN* modified to account for the cylindrical geometry of the detector[80,81]. The program *LSCALE*[82] was used to determine the wavelength-normalization curve using the intensities of symmetry-equivalent reflections measured at different wavelengths. No explicit absorption corrections were applied. The data were merged and scaled using *SCALA*[83]. Neutron data collection can be found in Supplementary Table 1.

**Joint X-ray/neutron (XN) refinement**. The joint XN structure of the deuterated M$^{pro}$/BBH-1 complex was refined using *nCNS*[84,85] and manual structure manipulation using COOT[70,71]. Initial rigid-body refinement was followed by several cycles of positional, atomic displacement parameter, and occupancy refinement. Correctness of sidechain conformations, hydrogen bonding, and orientations of D$_2$O water molecules in the structure was determined from $mF_O-DF_C$ difference neutron scattering length density maps. The $2mF_O-DF_C$ and $mF_O-DF_C$ neutron scattering length density maps were then examined to determine the correct orientations of hydroxyl (Ser, Thr, Tyr), thiol (Cys), and ammonium (Lys) groups, and protonation states of the enzyme residues. The protonation states of some disordered side chains on the protein surface could not be obtained directly and remained ambiguous. Water molecules were refined as D$_2$O. Water oxygen atoms were centered on their electron density peaks, and each molecule was rotated in accordance with the neutron scattering length density maps. Hydrogen positions in the protein were modeled as deuterium atoms because the protein was partially deuterated. Occupancies of D atoms were refined individually within the range of −0.56 (pure H) to 1.00 (pure D) because the neutron scattering length of H is −0.56 times that of D. Before depositing the neutron structure to the PDB, coordinates of a D atom were split into two records corresponding to an H and a D partially occupying the same site, both with positive partial occupancies that add up to unity. The percent D at a specific site is calculated according to the following formula: % D = {occupancy(D) + 0.56}/1.56. Neutron refinement statistics can be found in Supplementary Table 1.

**Isothermal titration calorimetry**. Purified M$^{pro}$ was diluted from a stock solution to 60 µM and dialyzed overnight at 4 °C against 25 mM Tris-HCl, pH 7.6, 20 mM NaCl and 1 mM TCEP (ITC buffer). The concentration of M$^{pro}$ was estimated based on its 280 nm absorbance. Stock solutions of inhibitors dissolved in DMSO were freshly diluted in ITC buffer prior to the titration and contained a final concentration not exceeding 1.5% DMSO. The protein solution was also adjusted to contain the same concentration of DMSO in ITC buffer. Titrations were performed with 30 µM M$^{pro}$ in the cell and 300 µM inhibitor at 28 °C on an iTC200 microcalorimeter (Malvern Instruments Inc., Westborough, MA). Control titrations of buffer with each of the inhibitors showed a negligible response. Data were processed and plots were generated using the Origin software provided with the instrument. For competitive inhibitors that bind at only one site, the dissociation constant ($K_d = 1/K_a$) is equivalent to the inhibition constant measured by enzyme kinetics ($K_i$).

**Antiviral assays**. Evaluation of antiviral activity of compounds BBH-1, BBH-2, NBH-2, and PF-7321332 was carried out in a 384-well plate with Vero E6 TMPRSS cells (purchased from Japanese Collection of Research Bioresources Cell Bank (JCRB), National Institutes of Biomedical Innovation, Health, and Nutrition) as described in Bocci et al.[86] using the USA-WA1/2020 (deposited by the Centers for Disease Control and Prevention and obtained through BEI Resources, NIAID, NIH, NR-52281). All activities with live virus were conducted at the UTHSC Regional Biocontainment Laboratory BSL-3. Compounds were evaluated in a dose-

response format starting at 10 micromolar and six additional twofold dilutions in duplicate. The cytotoxicity of compounds was tested either alone or in the presence of the P-glycoprotein inhibitor CP-100356. The antiviral activity was also assessed with the compound alone or in the presence of CP-100356 and SARS-CoV-2. CP-100356 was included at 2 µM concentration to assess whether our test compounds are being effluxed out of cells due to endogenous expression of P-glycoprotein in the cells line, which is known for PF-7321332[49]. Following incubation for 48 h at 5% CO$_2$ and 37 ºC, the percent cell viability was measured with CellTiterGlo. Signals were read with an EnVision® 2105 multimode plate reader. Cells alone (positive control) and cells plus virus (negative control) were set to 100% and 0% cell viability to normalize the data from the compound testing. The normalized data was used to calculate the 50% cytotoxic concentration (CC$_{50}$) and effective concentration (EC$_{50}$) using GraphPad version 9.3.1 by plotting the log(inhibitor) vs. normalized response with a variable slope (Hill = 1) (Table 2 and Supplementary Fig. 5).

Our 384-well HTS assay to measure antiviral activity in a dose-response format was validated as per HTS Assay Validation - Assay Guidance Manual - NCBI Bookshelf (https://www.ncbi.nlm.nih.gov/books/NBK83783/) and based on our prior HTS assays for single and dose-response screening validated for SARS-CoV-1[87]. The luminescent-based assay measures the inhibition of SARS-CoV-2-induced cytopathic effect (CPE) by measuring available ATP levels in cells with CellTiterGlo. The assay was validated in 384-well plates in the BSL-3 lab within the UTHSC Regional Biocontainment Laboratory. The assay is sensitive and robust, with Z values > 0.6, signal to background (S/B) > 20, and signal to noise (S/N) > 3. The assay is run with two replicates for each concentration, and the values are averaged and normalized to cells (positive control) and cells plus virus (negative control). The coefficient of variation of cells was 5.9 and for cells plus virus was 7.4 for the 384-well plate for the experiment for the compounds presented.

**Reporting summary**. Further information on research design is available in the Nature Research Reporting Summary linked to this article.

## Data availability
The data that support this study are available from the corresponding authors upon reasonable request. The structure and corresponding structure factors have been deposited into the protein data bank with the PDB accession codes 7TDU for M$^{pro}$/BBH-1, 7TEH for M$^{pro}$/BBH-2, 7TFR for Mpro/NBH-2, and 7SI9 for M$^{pro}$/PF-07321332. Source data are provided with this paper.

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

## Acknowledgements

This research used resources at the Center for Nanophase Materials Sciences, the Spallation Neutron Source, and the High Flux Isotope Reactor, which are DOE Office of Science User Facilities operated by the Oak Ridge National Laboratory. The Office of Biological and Environmental Research supported research at ORNL's Center for Structural Molecular Biology (CSMB), a DOE Office of Science User Facility. This research used resources of the Spallation Neutron Source Second Target Station Project at Oak Ridge National Laboratory (ORNL). ORNL is managed by UT-Battelle LLC for DOE's Office of Science, the single largest supporter of basic research in the physical sciences in the United States. The authors thank the Institut Laue Langevin (beamline LADI-DALI) for awarded neutron beamtime. We thank Dr. Hugh M. O'Neill from ORNL for assistance during the expression of the partially deuterated protein. We thank Annie Aniana from the National Institute of Diabetes and Digestive and Kidney Diseases (NIDDK) for excellent technical assistance. This work was also supported by the Intramural Research Program of NIDDK, NIH.

## Author contributions

D.W.K., L.C., P.V.B., and A.K. designed the study. G.P. and Q.Z. expressed the protein. D.W.K. purified the protein. K.L.W. performed protein deuteration. D.W.K. and A.K. crystallized the protein and inhibitor complexes. D.W.K., L.C., and A.K. collected X-ray data, reduced the data, and refined the structures. H.L. and P.V.B. synthesized compounds. J.M.L. produced a second source of M^pro and performed ITC experiments. M.A.A. collected mass spectrometry data. C.B.J., S.S., and J.P. performed antiviral assays. M.P.B. collected and reduced neutron diffraction data. D.W.K., H.L., P.V.B., and A.K. wrote the paper with help from all co-authors.

## Competing interests

The authors declare no competing interests.
