## [Peer Review File · Nature Communications]

Covalent narlaprevir- and boceprevir-derived hybrid inhibitors of SARS-CoV-2 main proteaseReviewers' Comments:

Reviewer #1:

Remarks to the Author:

In this article, Kneller et al. reported the design of three reversible covalent inhibitors for SARS-CoV-2 Mpro, including BBH-1, BBH-2 and NBH-2. These three compounds are hybrids of existing drugs PF-07321332 and boceprevir/narlaprevir, which are approved drugs with known mechanisms of action. Next, the authors determined the Kds of BBH-2, NBH-2, GC376 and PF-07321332 using ITC. Subsequently, the authors determined the Xray structures of Mpro in complex with BBH1, BBH-2 and NHB-2 at room temp. and/or 100K, as well as the neutron structure of Mpro/BBH-1. They also tested the in cyto potency and cyto-toxicity for these new compounds on Vero-E6 cells. The introduction section provided a thorough analysis of the discovery of the SARS-CoV-2 Mpro inhibitors.

From the perspective of drug design, the warhead and the P1-P4 part of the new compounds are directly from known parent compounds with antiviral activities. Therefore, it is foreseeable that the hybrids obtained by combining the two parts would be active. Meanwhile, the chemical structures of the new compounds are not much different from PF-07321332, so the chemical structures of these three compounds are not significantly novel.

Compared with PF-07321332, the Kd and antiviral activity EC50 of the compounds BBH-1/BBH-2/NBH-2 were even weaker, with the best EC50 of BBH-2 being 0.88 μ M, while the EC50 of PF-07321332 being 0.25 μ M.

The neutron structure of Mpro/BBH-1 provided some new information. It helped the authors see the protonation states of some ionizable residues of Mpro, which revealed some novel interactions.

This study would be more meaningful if the new compounds can perform well in terms of druggability (e.g. pharmacokinetics).

Major issues:

1. It is mentioned in lines 228-229 that the carbonyl group at the P4 position in the compound BBH-1 has an unconventional C-H ...O hydrogen bond interaction with Gln189 on the side chain, but this is not seen in any figure. In addition, the reference (Owen et al., Science, 2021) provided detailed analysis of regular (conventional) hydrogen bonds and electrostatic interactions for the side chain of Gln189 and PF-07321332 in fig. 2 of (Owen, Science, 2021). How do the authors know that the unconventional C-H ...O bond may account for the favorable enthalpy of binding and high potency of PF-07321332? Or, how much does the unconventional C-H...O bond contribute to enthalpy and Kd in comparison with regular H-bond?

2. Is there any isomer products in the final condensation reaction in the synthesis of BBH-1/BBH-2/NBH-2? If so, how was the isolation performed and how was the correct configuration of the products determined?

3. The authors mentioned "specific" inhibitors multiple times throughout the manuscript, e.g. in line 158 "these compounds are viable for further design of specific antivirals against SARS-CoV-2.". The authors may need to test the specificity of these compounds against SARS-CoV, MERS-CoV and SARS-CoV-2 Mpros before claiming this. For example, Owen et al. in Science, 2021 have done such a comparison in Fig. 3B.

4. Fig S5, Cyto-toxicity and antiviral activities.....: the labelling on the x-axis is wrong. If Log10(A)=10, then A=10¹⁰ μ M? Also, If the authors extracted EC50 values based on these data, it is better to provide fitting curves instead of connecting lines. In table 2, can the authors provide standard errors or standard deviations for EC50 values?

Minor issues:

1. In line 155, BBH-2 and NBH-2 have Kds of 26-30 nM, they are weaker than PF-073-21332 with a Kd of 7 nM, so they are not "favorable" in comparison.
2. In line 584, antiviral experiments were done in duplicates. Triplicates will be better to reveal the true values. In line 590, it will be helpful if the authors give full details of "CellTiterGlo" to people who are not familiar with this technique.
3. As shown in table 2, the EC50s of BBH-1, BBH-2 and NBH-2 were 16.1 μ M, 15.4 μ M and 13.9 μ M, respectively, which were not significantly different from the EC50 values of boceprevir and narlaprevir reported in the literature 2, 3. It is recommended to add boceprevir and narlaprevir as controls in antiviral activity experiments.
4. In line 372-374, "This led us to the design of hybrid covalent inhibitors BBH-1, BBH-2 and NBH-2 with a 100-fold improved affinity to Mpro compared to that of boceprevir and narlaprevir". The manuscript did not give sufficient data to support this conclusion. Boceprevir and narlaprevir are not included in the ITC experiments in Table 1, and there are no ITC data of boceprevir and narlaprevir in the three literatures cited by the author. The authors may check ref 1 by Bai et al. for ITC Kds of boceprevir and narlaprevir (21 μ M and 82 μ M).

References:

1. Bai, Y.; Ye, F.; Feng, Y.; Liao, H.; Song, H.; Qi, J.; Gao, G. F.; Tan, W.; Fu, L.; Shi, Y., Structural basis for the inhibition of the SARS-CoV-2 main protease by the anti-HCV drug narlaprevir. *Signal Transduct Target Ther* 2021, 6 (1), 51.
2. Fu, L.; Ye, F.; Feng, Y.; Yu, F.; Wang, Q.; Wu, Y.; Zhao, C.; Sun, H.; Huang, B.; Niu, P.; Song, H.; Shi, Y.; Li, X.; Tan, W.; Qi, J.; Gao, G. F., Both Boceprevir and GC376 efficaciously inhibit SARS-CoV-2 by targeting its main protease. *Nat Commun* 2020, 11 (1), 4417.
3. Ma, C.; Sacco, M. D.; Hurst, B.; Townsend, J. A.; Hu, Y.; Szeto, T.; Zhang, X.; Tarbet, B.; Marty, M. T.; Chen, Y.; Wang, J., Boceprevir, GC-376, and calpain inhibitors II, XII inhibit SARS-CoV-2 viral replication by targeting the viral main protease. *Cell Res* 2020, 30 (8), 678-692.

Reviewer #2:

Remarks to the Author:

The work by Kneller and colleagues entitled "Covalent narlaprevir- and boceprevir-derived hybrid inhibitors of SARS-CoV-2 main protease: room-temperature X-ray and neutron crystallography, binding thermodynamics, and antiviral activity" builds on previous work in this area (1-7) and presents three reversible covalent inhibitors (BBH-1, BBH-2 and NBH-2) derived from two hepatitis C protease inhibitors. The authors solved the joint neutron/X-ray RT structure of the viral Mpro protease in complex with BBH-1 and the RT X-ray structure of Mpro in complex with the other two compounds. Binding thermodynamics was assessed by ITC and antiviral activity investigated using Vero E6 TMPRSSS cells as described in Bocci et al. (2020).

From a technical standpoint the work is sound although the antiviral assay seems be a bit limited in its scope. As they stand the antiviral activity BBH-1, BBH-2 and NBH-2 is lower than that of the FDA-approved inhibitor PF-07321332. Clearly, the compounds presented are a useful development and highlight some interesting features in terms interaction with the Mpro cysteine protease, however, I am not convinced that this work represents a sufficient advancement in the field to warrant its

publication in NCOMMS.

1 Kneller, D. W.; Galanie, S.; Phillips, G.; O'Neill, H. M.; Coates, L.; Kovalevsky, A. Malleability of the SARS-CoV-2 3CL Mpro Active-Site Cavity Facilitates Binding of Clinical Antivirals. *Structure* 2020a, 28, 1313–1320.

2 Kneller, D. W., Phillips, G., Weiss, K. L., Pant, S., Zhang, Q., O'Neill, H. M., Coates, L., and Kovalevsky, A. (2020b) Unusual zwitterionic catalytic site of SARS-CoV-2 main protease revealed by neutron crystallography. *J. Biol. Chem.* 295, 17365–17373.

3 Kneller, D. W., Phillips, G., O'Neill, H. M., Jedrzejczak, R., Stols, L., Langan, P., Joachimiak, A., Coates, L., and Kovalevsky, A. (2020c) Structural plasticity of SARS-CoV-2 3CL Mpro active site cavity revealed by room temperature X-ray crystallography. *Nat. Commun.* 11, 3202.

4 Kneller, D. W., Phillips, G., Kovalevsky, A., and Coates, L. (2020d) Room-temperature neutron and X-ray data collection of 3CL Mpro from SARS-CoV-2. *Acta Crystallogr. Sect. F Struct. Biol. Commun.* 76, 483–487.

5 D.W. Kneller, Q. Zhang, L. Coates, J.M. Louis, A. Kovalevsky (2021a) Michaelis-like complex of SARS-CoV-2 main protease visualized by room-temperature X-ray crystallography. *IUCR J.* 8, 973-979.

6 D.W. Kneller, H. Li, S. Galanie, G. Phillips, A. Labbe, K.L. Weiss, Q. Zhang, M.A. Arnould, A. Clyde, H. Ma, A. Ramanathan, C.B. Jonsson, M.S. Head, L. Coates, J. M. Louis, P.V. Bonnesen, A. Kovalevsky (2021b) Structural, electronic and electrostatic determinants for inhibitor binding to subsites S1 and S2 in SARS-CoV-2 main protease. *J. Med. Chem.* 64, 17366-17383.

7 D.W. Kneller, G. Phillips, K.L. Weiss, Q. Zhang, L. Coates, A. Kovalevsky (2021c) Direct observation of protonation state modulation in SARS-CoV-2 main protease upon inhibitor binding with neutron crystallography. *J. Med. Chem.* 64, 4991-5000.

Reviewer #3:

Remarks to the Author:

This is an excellent paper using neutron and X-ray crystallography (complemented with molecular biophysics measures of binding (ITC) and cell-based antiviral activity experiments) to provide the details of small molecule inhibition of SARS-CoV-2 main protease. With neutron crystallography the precise configuration of protons is given and thereby the charge state of the active site. It is very interesting how the protonation state changes with SMI binding. This article will be of interest to many scientists because of the power of the research tools used and the importance of the problem to the world, in particular as we battle the variants. Suggestions for the future design of inhibitors is concluded. I have the following comment to improve the presentation.

The abstract needs some work, there are 3 sentences that are too vague.

- In the 3rd line it says the SMI are important "that is subject to challenges". What does this mean, I think they mean to say that the SMI has advantages over other therapeutics, such as spike-based vaccines, that are effected by mutations in variants. Please rewrite and make this sentence more descriptive of what you mean.

-abstract line 7, this sentence is way too long (6 lines) and the phrase "were derived from" isn't very clear. This should probably be two sentences and an indication of the uniqueness of how the hybrid SMIs were redesigned by splicing together components to make a hybrid molecule.

- abstract line 13, "unconventional interactions", what are these? Intriguing, yet frustrating to the

reader. Please explain.

Introduction need some work:

- The paragraph encompassing page 4 needs a figure of the active site, either with pymol or chemdraw, so that the description of the enzyme can be better understood by a broader audience (other than main protease experts).
- The virus targeted by Narlaprevir should be described.
- As it was used in the redesign of SMIs, Telaprevir should be in Figure 1. I would put Boceprevir, Narlaprevir, Telaprevir on one line; GC-376 and PF-07321332 on the next line and BBH-1, BBH-2 and NBH-2 on the third line of the figure. Please label P1-P5(orP4) for all the SMI to help in the interpretation of later figures where these labels are used.

Figure 2 part D the protein bonds involving carbon should be pink to match the protein ribbon.

Figure 5, please put angstrom symbol after the distances.

In the crystallization write up, please include reservoir volumes as well as drop volumes.

In the supplement Figure S5, please put a little more description of the experiment in the figure legend so that it stands alone, for example what kind of cells are used. Does this data correspond with Table 2?

REVIEWER COMMENTS

Reviewer #1:

In this article, Kneller et al. reported the design of three reversible covalent inhibitors for SARS-CoV-2 Mpro, including BBH-1, BBH-2 and NBH-2. These three compounds are hybrids of existing drugs PF-07321332 and boceprevir/narlaprevir, which are approved drugs with known mechanisms of action. Next, the authors determined the Kds of BBH-2, NBH-2, GC376 and PF-07321332 using ITC. Subsequently, the authors determined the Xray structures of Mpro in complex with BBH1, BBH-2 and NHB-2 at room temp. and/or 100K, as well as the neutron structure of Mpro/BBH-1. They also tested the in cyto potency and cyto-toxicity for these new compounds on Vero-E6 cells. The introduction section provided a thorough analysis of the discovery of the SARS-CoV-2 Mpro inhibitors.

From the perspective of drug design, the warhead and the P1-P4 part of the new compounds are directly from known parent compounds with antiviral activities. Therefore, it is foreseeable that the hybrids obtained by combining the two parts would be active. Meanwhile, the chemical structures of the new compounds are not much different from PF-07321332, so the chemical structures of these three compounds are not significantly novel.

Compared with PF-07321332, the Kd and antiviral activity EC50 of the compounds BBH-1/BBH-2/NBH-2 were even weaker, with the best EC50 of BBH-2 being 0.88 μ M, while the EC50 of PF-07321332 being 0.25 μ M.

The neutron structure of Mpro/BBH-1 provided some new information. It helped the authors see the protonation states of some ionizable residues of Mpro, which revealed some novel interactions.

This study would be more meaningful if the new compounds can perform well in terms of druggability (e.g. pharmacokinetics).

Response: We appreciate the reviewer's comments. The introduction already includes text regarding our design strategy: "Building from our recent hepatitis C virus protease inhibitor repurposing work (Kneller et al., 2020a), we have followed an analogous strategy, working independently from and in parallel with the other research groups, to design hybrid reversible covalent inhibitors of M^{pro} that combine the chemical features of boceprevir and narlaprevir (ArlansaTM, R-Pharm, Russia, Arasappan et al., 2010; Isakov et al., 2016) with earlier SARS-CoV-1 M^{pro} inhibitors containing the widely used P1 γ -lactam and electrophilic arylketone or nitrile functionalities as warheads (Thanigaimalai et al., 2013; Konno et al., 2013; Chuck et al., 2013). Our design logic also considered the observations and conclusions from our recent neutron structure of M^{pro} in complex with another hepatitis C virus protease inhibitor, telaprevir (Kneller et al., 2021c)."

Major issues:

1. It is mentioned in lines 228-229 that the carbonyl group at the P4 position in the compound BBH-1 has an unconventional C-H...O hydrogen bond interaction with Gln189 on the side chain, but this is not seen in any figure. In addition, the reference (Owen et al., Science, 2021) provided detailed analysis of regular (conventional) hydrogen bonds and electrostatic interactions for the side chain of Gln189 and PF-07321332 in fig. 2 of (Owen, Science, 2021). How do the authors know that the unconventional C-H...O bond may account for the favorable enthalpy of binding and high potency of PF-07321332? Or, how much does the unconventional C-H...O bond contribute to enthalpy and Kd in comparison with regular H-bond?

Response: We included the C-H...O interaction in a revised Figure 2d. The contribution of C-H...O hydrogen bonds to biological structure and function has been extensively studied (Derewenda et al. *J. Mol. Biol.* 1994, 241, 83-93; Fabiola et al. *Acta Cryst.* 1997, D53, 316-320; Jiang & Lai *J. Biol. Chem.* 2002, 277, 37732-37740; Desiraju *ChemComm.* 2005, 2995-3001; Horowitz & Trievel *J. Biol. Chem.* 2012, 287, 41576-41582). The energy of this nonconventional interaction is ~0.5-4 kcal/mol, depending on the acidity of the C-H bond, the basicity of the oxygen atom and the geometry of the interaction (Gu et al. *J. Am. Chem. Soc.* 1999, 121, 9411-9422). We have added relevant citations to the manuscript. However, it is very hard to predict how much favorable enthalpy the C-H...O hydrogen bond between BBH-1 and Mpro would contribute to the inhibitor binding. Of note, one cannot simply predict how much energetically the conventional hydrogen bonds contribute to the inhibitor binding either. To answer this question would require high-level computational studies that are beyond the scope of our manuscript.

2. Is there any isomer products in the final condensation reaction in the synthesis of BBH-1/BBH-2/NBH-2? If so, how was the isolation performed and how was the correct configuration of the products determined?

Response: We addressed the rotamer issue with BBH-1 in the SI, with a reference to the Owen et al., 2021 paper on that point. The nitrile warheads do not have isomer/rotamer issues, as is also noted in the cited paper (no isomer issue identified for PF-07321332). Isolation was performed by column chromatography and all inhibitors were fully characterized by proton and carbon NMR, and MALDI-ToF MS (described in the SI).

3. The authors mentioned “specific” inhibitors multiple times throughout the manuscript, e.g. in line 158 “these compounds are viable for further design of specific antivirals against SARS-CoV-2.”. The authors may need to test the specificity of these compounds against SARS-CoV, MERS-CoV and SARS-CoV-2 Mpros before claiming this. For example, Owen et al. in Science, 2021 have done such a comparison in Fig. 3B.

Response: We removed the word “specific” from the Results, as testing specificity of the compounds on several other Mpros may require several months of significant time and effort.

4. Fig S5, Cyto-toxicity and antiviral activities.....: the labelling on the x-axis is wrong. If Log10(A)=10, then A=10¹⁰ uM? Also, If the authors extracted EC50 values based on these data, it is better to provide fitting curves instead of connecting lines. In table 2, can the authors provide standard errors or standard deviations for EC50 values?

Response: The figure has been revised to include only the fitting curves. The x-axis was relabeled. As this is a HTS assay qualified to be run in single concentrations, we use the plate as our statistical evaluation of the data and not the individual data points. However, we are running compounds with two biological replicates and two technical replicates per plate, and so this is included in Table 2 as 95% CI values. The Z value, coefficient of variation, S/N and S/B are now provided in the materials and methods.

Minor issues:

1. In line 155, BBH-2 and NBH-2 have Kds of 26-30 nM, they are weaker than PF-073-21332 with a Kd of 7 nM, so they are not “favorable” in comparison.

Response: We agree with the reviewer. In fact, we acknowledge in the section “Binding thermodynamics of the inhibitors” that PF-07321332 demonstrates “superior binding affinity compared to the other two inhibitors”.

2. In line 584, antiviral experiments were done in duplicates. Triplicates will be better to reveal the true values. In line 590, it will be helpful if the authors give full details of “CellTiterGlo” to people who are not familiar with this technique.

Response: We have included in the Materials and Methods the following details of the HTS assay used to measure antiviral activity and a citation: “Our 384-well HTS assay to measure antiviral activity in a dose response format was validated as per [HTS Assay Validation - Assay Guidance Manual - NCBI Bookshelf \(https://www.ncbi.nlm.nih.gov/books/NBK83783/\)](https://www.ncbi.nlm.nih.gov/books/NBK83783/) and based on our prior HTS assays for single and dose response screening validated for SARS-CoV-1 (Severson et al. 2006). The luminescent-based assay measures the inhibition of SARS-CoV-2-induced cytopathic effect (CPE) by measuring available ATP levels in cells with CellTiterGlo. The assay was validated in 384-well plates in a BSL3 containment facility. The assay is sensitive and robust, with Z values > 0.6, signal to background (S/B) > 20, and signal to noise (S/N) > 3. The assay is run with two replicates for each concentration, and the values are averaged and normalized to cells (positive control) and cells plus virus (negative control). The coefficient of variation of cells was 5.9 and for cells plus virus was 7.4 for the 384-well plate for the experiment for the compounds presented.”

3. As shown in table 2, the EC50s of BBH-1, BBH-2 and NBH-2 were 16.1 μ M, 15.4 μ M and 13.9 μ M, respectively, which were not significantly different from the EC50 values of boceprevir and narlaprevir reported in the literature 2, 3. It is recommended to add boceprevir and narlaprevir as controls in antiviral activity experiments.

Response: We acknowledge that the antiviral activity of the hepatitis C virus protease inhibitors has been established previously (Ma et al. Cell Research 2020; Fu et al. Nat. Commun. 2020). Unfortunately, we are unable to perform such antiviral assays on boceprevir and narlaprevir in a timely manner because we exhausted our local supply and the lead time to procure these chemicals and carry out the assays will take several months. We therefore cite the two above-mentioned publications.

4. In line 372-374, “This led us to the design of hybrid covalent inhibitors BBH-1, BBH-2 and NBH-2 with a 100-fold improved affinity to Mpro compared to that of boceprevir and narlaprevir”. The manuscript did not give sufficient data to support this conclusion. Boceprevir and narlaprevir are not included in the ITC experiments in Table 1, and there are no ITC data of boceprevir and narlaprevir in the three literatures cited by the author. The authors may check ref 1 by Bai et al. for ITC Kds of boceprevir and narlaprevir (21 μ M and 82 μ M).

Response: We thank the reviewer for pointing this out. We did attempt ITC analyses with boceprevir and narlaprevir under the same conditions as shown for compounds listed in Table 1 for comparison prior to our original submission. However, we saw no thermal response, meaning that these inhibitors are weak binders with K_d 's likely in the low micromolar range. We refrain from citing ITC data from Bai et al., (<https://www.nature.com/articles/s41392-021-00468-9>) because the conditions of their ITC analysis include 10% DMSO. We don't exceed 1.5% DMSO in our measurements even when matching the cell and syringe solutions with the same concentration of DMSO. In recent studies we show the K_d determined by ITC for GC376 binding to MPro^{WT} is 0.15 μ M (Nashed et al., Fig. 7). This points to a 3-fold weaker binding in the presence of 10% DMSO for GC376 to MPro^{WT}. For this reason, we now indicate the range of IC_{50} values obtained with these inhibitors citing references for comparison with our compounds. We revised the following sentence in the Discussion to read: ” This led us to the design of hybrid covalent inhibitors BBH-1, BBH-2 and NBH-2 with >100-fold improved affinity to M^{pro}, when compared to the IC_{50} values of boceprevir (3.1-8.0 μ M) and narlaprevir (4.7-5.1 μ M) determined previously (Ma et al., 2020; Fu et al., 2020; Kneller et al. 2020a)...”

A side-by-side estimation of MPro^{WT} inhibition by boceprevir and GC376 show IC_{50} values of $8 \pm 1.5 \mu$ M and $0.15 \pm 0.03 \mu$ M (Fu et al., <https://www.nature.com/articles/s41467-020-18233-x>), respectively, where the IC_{50} value for GC376 exactly matches our estimate by ITC. For narlaprevir, we use a side-by-side IC_{50} estimation of boceprevir and narlaprevir showing values of 4.13 and 5.73 μ M, respectively (<https://www.sciencedirect.com/science/article/pii/S2211383521004299>), where the IC_{50} value obtained for boceprevir is 2-fold lower than estimated by Fu et al. Our own data for boceprevir and narlaprevir (Kneller et al., Structure 2020) show IC_{50} values of 3.1 μ M (CI [2.8, 3.5]), and 5.1 μ M (CI [4.3, 6.1]), respectively.

References:

1. Bai, Y.; Ye, F.; Feng, Y.; Liao, H.; Song, H.; Qi, J.; Gao, G. F.; Tan, W.; Fu, L.; Shi, Y., Structural basis for the inhibition of the SARS-CoV-2 main protease by the anti-HCV drug narlaprevir. *Signal Transduct Target Ther* 2021, 6 (1), 51.
2. Fu, L.; Ye, F.; Feng, Y.; Yu, F.; Wang, Q.; Wu, Y.; Zhao, C.; Sun, H.; Huang, B.; Niu, P.; Song, H.; Shi, Y.; Li, X.; Tan, W.; Qi, J.; Gao, G. F., Both Boceprevir and GC376 efficaciously inhibit SARS-CoV-2 by targeting its main protease. *Nat Commun* 2020, 11 (1), 4417.
3. Ma, C.; Sacco, M. D.; Hurst, B.; Townsend, J. A.; Hu, Y.; Szeto, T.; Zhang, X.; Tarbet, B.; Marty, M. T.; Chen, Y.; Wang, J., Boceprevir, GC-376, and calpain inhibitors II, XII inhibit SARS-CoV-2 viral replication by targeting the viral main protease. *Cell Res* 2020, 30 (8), 678-692.

Reviewer #2:

The work by Kneller and colleagues entitled "Covalent narpilaprevir- and boceprevir-derived hybrid inhibitors of SARS-CoV-2 main protease: room-temperature X-ray and neutron crystallography, binding thermodynamics, and antiviral activity" builds on previous work in this area (1-7) and presents three reversible covalent inhibitors (BBH-1, BBH-2 and NBH-2) derived from two hepatitis C protease inhibitors. The authors solved the joint neutron/X-ray RT structure of the viral Mpro protease in complex with BBH-1 and the RT X-ray structure of Mpro in complex with the other two compounds. Binding thermodynamics was assessed by ITC and antiviral activity investigated using Vero E6 TMPRSSS cells as described in Bocci et al. (2020).

From a technical standpoint the work is sound although the antiviral assay seems to be a bit limited in its scope. As they stand the antiviral activity of BBH-1, BBH-2 and NBH-2 is lower than that of the FDA-approved inhibitor PF-07321332. Clearly, the compounds presented are a useful development and highlight some interesting features in terms of interaction with the Mpro cysteine protease, however, I am not convinced that this work represents a sufficient advancement in the field to warrant its publication in NCOMMS.

1 Kneller, D. W.; Galanie, S.; Phillips, G.; O'Neill, H. M.; Coates, L.; Kovalevsky, A. Malleability of the SARS-CoV-2 3CL Mpro Active-Site Cavity Facilitates Binding of Clinical Antivirals. *Structure* 2020a, 28, 1313–1320.

2 Kneller, D. W., Phillips, G., Weiss, K. L., Pant, S., Zhang, Q., O'Neill, H. M., Coates, L., and Kovalevsky, A. (2020b) Unusual zwitterionic catalytic site of SARS-CoV-2 main protease revealed by neutron crystallography. *J. Biol. Chem.* 295, 17365–17373.

3 Kneller, D. W., Phillips, G., O'Neill, H. M., Jedrzejczak, R., Stols, L., Langan, P., Joachimiak, A., Coates, L., and Kovalevsky, A. (2020c) Structural plasticity of SARS-CoV-2 3CL Mpro active site cavity revealed by room temperature X-ray crystallography. *Nat. Commun.* 11, 3202.

4 Kneller, D. W., Phillips, G., Kovalevsky, A., and Coates, L. (2020d) Room-temperature neutron and X-ray data collection of 3CL Mpro from SARS-CoV-2. *Acta Crystallogr. Sect. F Struct. Biol. Commun.* 76, 483–487.

5 D.W. Kneller, Q. Zhang, L. Coates, J.M. Louis, A. Kovalevsky (2021a) Michaelis-like complex of SARS-CoV-2 main protease visualized by room-temperature X-ray crystallography. *IUCR J.* 8, 973-979.

6 D.W. Kneller, H. Li, S. Galanie, G. Phillips, A. Labbe, K.L. Weiss, Q. Zhang, M.A. Arnould, A. Clyde, H. Ma, A. Ramanathan, C.B. Jonsson, M.S. Head, L. Coates, J. M. Louis, P.V. Bonnesen, A. Kovalevsky (2021b) Structural, electronic and electrostatic determinants for inhibitor binding to subsites S1 and S2 in SARS-CoV-2 main protease. *J. Med. Chem.* 64, 17366-17383.

7 D.W. Kneller, G. Phillips, K.L. Weiss, Q. Zhang, L. Coates, A. Kovalevsky (2021c) Direct observation of protonation state modulation in SARS-CoV-2 main protease upon inhibitor binding with neutron crystallography. *J. Med. Chem.* 64, 4991-5000.

Response: We thank the reviewer for careful consideration of our manuscript. We believe that the results presented in our manuscript are of timely importance for the drug design efforts targeting SARS-CoV-2 protease as we report room-temperature neutron and X-ray crystal

structures that provide unique atomic information about inhibitor binding, ITC measurements mostly lacking in the literature for this enzyme that provide thermodynamics of binding, and the designed compounds are antivirally active providing insights into further design of Mpro inhibitors.

Reviewer #3:

This is an excellent paper using neutron and X-ray crystallography (complemented with molecular biophysics measures of binding (ITC) and cell-based antiviral activity experiments) to provide the details of small molecule inhibition of SARS-CoV-2 main protease. With neutron crystallography the precise configuration of protons is given and thereby the charge state of the active site. It is very interesting how the protonation state changes with SMI binding. This article will be of interest to many scientists because of the power of the research tools used and the importance of the problem to the world, in particular as we battle the variants. Suggestions for the future design of inhibitors is concluded. I have the following comment to improve the presentation.

Response: We thank the reviewer for their enthusiasm towards our manuscript, which we share. We greatly appreciate the reviewer's time and effort to assess our work.

The abstract needs some work, there are 3 sentences that are too vague.
- In the 3rd line it says the SMI are important "that is subject to challenges". What does this mean, I think they mean to say that the SMI has advantages over other therapeutics, such as spike-based vaccines, that are effected by mutations in variants. Please rewrite and make this sentence more descriptive of what you mean.

Response: We revised this sentence to read: "Small-molecule antivirals can provide an important therapeutic treatment option that has advantages over other therapeutics, such as vaccines and therapeutic antibodies targeting the viral spike protein."

-abstract line 7, this sentence is way too long (6 lines) and the phrase "were derived from" isn't very clear. This should probably be two sentences and an indication of the uniqueness of how the hybrid SMIs were redesigned by splicing together components to make a hybrid molecule.

Response: We did not find a 6-line long sentence in the Abstract. However, we revised the following sentence to address the reviewer's comment: "We performed the design and characterization of three reversible covalent hybrid inhibitors BBH-1, BBH-2 and NBH-2. These structures were created by splicing components of hepatitis C protease inhibitors boceprevir and narlaprevir, and known SARS-CoV-1 protease inhibitors."

- abstract line 13, "unconventional interactions", what are these? Intriguing, yet frustrating to the reader. Please explain.

Response: The reference to unconventional F...O interactions is added in the sentence.

Introduction need some work:

- The paragraph encompassing page 4 needs a figure of the active site, either with pymol or chemdraw, so that the description of the enzyme can be better understood by a broader audience (other than main protease experts).

Response: We have introduced a figure orienting the audience to the active site of SARS-CoV-2 Mpro.

- The virus targeted by Narlaprevir should be described.

Response: Narlaprevir is a hepatitis C virus protein inhibitor. We added this reference in the Introduction now.

- As it was used in the redesign of SMIs, Telaprevir should be in Figure 1. I would put Boceprevir, Narlaprevir, Telaprevir on one line; GC-376 and PF-07321332 on the next line and BBH-1, BBH-2 and NBH-2 on the third line of the figure. Please label P1-P5(orP4) for all the SMI to help in the interpretation of later figures where these labels are used.

Response: We did not use telaprevir for designing the hybrid inhibitors. Figure 1 has been revised as recommended.

Figure 2 part D the protein bonds involving carbon should be pink to match the protein ribbon.

Response: We tried to color protein bonds in Figure 2D in pink. That, however, makes it hard to see protein residues. We therefore decided to keep the original color pallet.

Figure 5, please put angstrom symbol after the distances.

Response: We revised the figure accordingly.

In the crystallization write up, please include reservoir volumes as well as drop volumes.

Response: We have included this data in the manuscript now.

In the supplement Figure S5, please put a little more description of the experiment in the figure legend so that it stands alone, for example what kind of cells are used. Does this data correspond with Table 2?

Response: We added more description into the Figure S5 legend.